# Stabilization/Solidification of Heavy Metals and PHe Contaminated Soil with β-Cyclodextrin Modified Biochar (β-CD-BC) and Portland Cement

**DOI:** 10.3390/ijerph19031060

**Published:** 2022-01-18

**Authors:** Geng Li, Haibo Li, Yinghua Li, Xi Chen, Xinjing Li, Lixin Wang, Wenxin Zhang, Ying Zhou

**Affiliations:** School of Resources and Civil Engineering, Northeastern University, 3-11 Wenhua Road, Heping District, Shenyang 110819, China; 1970915@stu.neu.edu.cn (G.L.); chenxineu@mail.neu.edu.cn (X.C.); 2001049@stu.neu.edu.cn (X.L.); 1970919@stu.neu.edu.cn (L.W.); 1901077@stu.neu.edu.cn (W.Z.); 20192118@stu.neu.edu.cn (Y.Z.)

**Keywords:** β-CD modified BC, stabilization/solidification, response surface methodology, synchronous adsorption investigations, PHe

## Abstract

Conventional stabilization/solidification materials have defects in the simultaneous treatment of heavy metals (HMs) and phenanthrene (PHe). In order to solve this problem, a new functional material β-cyclodextrin modified biochar (β-CD-BC) was prepared by integrating the properties of biochar (BC) and the hydrophilic and hydrophobic properties of the β-CD surface and combined with Portland cement (PC) to cure and stabilize HMs and PHe. The effect of key parameters on the treatment effect was discussed by response surface method. The results showed that the minimum leaching concentration if HMs was 16.81 mg·L^−1^, and the leaching concentration of PHe can be as low as 0.059 μg/kg under the conditions of β-CD-BC and Portland cement ratio of 9.75% and 11.4%, curing for 22.85 d. The weak acid soluble state reduced from 9~13% to 0.5~6%, the residual state was increased from 37~61% to 77~87%. The unconfined compressive strength of sample is more than 50 kPa. The results of this study can provide a new technical scheme for long-term curing and stabilization of HMs and PHe.

## 1. Introduction

Heavy metals (HMs) and phenanthrene (PHe) widely exist in the natural environment [1], which can not only accumulate in the human body through skin and respiratory tract and endanger human health, but also migrate in the soil and causing human water risk [2]. The research and development of soil remediation technology have not attracted enough attention. Therefore, the study is of great significance for the solidification/stabilization (S/S) of HMs and PHe in soil.

At present, the commonly used treatment technologies of HMs and PHe contaminated soil include S/S [3], leaching [4], combined Phyto microbial remediation [5,6], and thermal desorption [7]. In the United States, 80% of the S/S projects of Superfund use cement as the curing agent. S/S technology has attracted much attention due to its advantages of simple operation, low cost, and good treatment effect. The selection of repair materials for this technology is the key. At present, common S/S materials at home and abroad include cement, lime, organic matter, biochar (BC), montmorillonite, fly ash, etc. [8]. These materials reduce the mobility and bioavailability of HMs by physical and chemical reactions such as adsorption, precipitation, redox, and ion exchange with HMs [9]. The latest research mainly focuses on composite materials made from the above materials. The study found that using fly ash, quicklime, and blast furnace slag as composite materials can significantly stabilize HMs such as Zn, Pb, Cu, and Cr in contaminated soil [10]. Based on X-ray diffraction, TGA, and SEM tests, Sora et al. [11] proved that organic pollutants can affect the type and quantity of cement hydration products and make the solidified body present more internal pores. Wang et al. [12] found that fly ash can improve the leaching toxicity of cement solidified organic pollutants. Still, it has a weak fixation effect on HMs, which proves that the fixation mechanism of organic matter and HMs is different. Kogbara et al. found that increasing the cement content can enhance the fixation of HMs, but has no apparent effect on TPH [13].

BC is an environmentally friendly material, cheap, and has many oxygen functional groups, large surface area, and well-developed pore structure [14]. It has the ability and potential to combine with different types of pollutants. BC can stabilize HMs, but it is feeble in stabilizing PHe. β-cyclodextrin (β-CD) could simultaneously achieve efficient cleanup of PHe and Cr, Cd, Cu, Pb, and Zn through avoiding the competitive behaviors between them, which were due to the different adsorption for Cr, Cd, Cu, Pb, and Zn (i.e., electrostatic attraction and complexation) and PHe (i.e., host–guest supramolecular and π-π interactions) [15,16,17]. Β-CD-BC can simultaneously solidify and stabilize HMs and Phe in soil.

To solve the above problems, BC, PC, and β-CD-BC use as S/S agents. Response surface optimization experiment was used to carry out an S/S experiment on heavy metal organic matter, to explore the influence of various factors on the leaching concentration of heavy metals and PHe, and to determine the optimal process parameters. The unconfined compressive strength of the solidified body was selected, and the mechanism of S/S was explored through the analysis of the morphological characteristics of HMs, to provide the basis for the risk control and remediation of the soil polluted by the heavy metal organic compounds.

## 2. Materials and Methods

### 2.1. Materials and Reagents

Chemical reagents including β-CD(CAS: 7585-39-9), epichlorohydrin (EPI)(CAS:106-89-8), HCl(CAS:7647-01-0), NaOH(CAS:1310-73-3) and Pb(NO_3_)_2_(CAS:1099-74-8), Cu_S_O_4_(CAS:7758-98-7), ZnCl_2_(CAS:7646-85-7), Cr(NO_3_)_3_(CAS:13548-38-4), Cd(NO_3_)_2_(CAS:10325-94-7) were all purchased from Xin Ke. (Shenyang, China). All reagents were of analytical grade. Rice husks were acquired in Shenyang, Liaoning province of China.

The soil sample was collected from the surface layer (0–30 cm) of an unpolluted calcareous soil at the College of Agriculture Shenyang. Table 1 presents physical and chemical properties of soil.

### 2.2. Preparation and Modification of Biochar

Pyrolysis produces of rice husk was carried out in atmosphere furnace(OFT-1200X-S, Hefei, China) under nitrogen (N_2_) flow of 0.2 L/min with a heating rate of 10 °C/min at 400 °C for 3 h. After the reactor was set to cool to room temperature, BC was obtained. Before dry preservation, BC was ground through a 100 mesh sieve.

Modification BC: As shown in Figure 1, 10.00 g of β-CD and 4.80 g of EPI in a molar ratio of 4:1 were added to 200 mL of 5% NaOH solution. The mixture was mixed for 4 h (320 rpm) at a temperature (25 °C) to afford the modification solution. Then, 5.00 g BC was weighed out and mixed with 100 mL of modification solution. The mixture was stirred at room temperature for 1.5 h (320 rpm), washed with deionized water, and filtered under suction several times, until the pH of BC was neutral. Then BC was left to stand overnight. The product was dried at 75 °C in an oven until its weight was stable, yielding β-CD functionalized biochar (β-CD-BC) [18].

### 2.3. Experimental Method

#### 2.3.1. Solidification/Stabilization Experiment

A single factor level experiment was designed to determine the factors and level of orthogonal experiment. To explore the influence of the addition amount of BC and β-CD-BC, 10% PC and 30% deionized water were added, cured in a 24 °C dry environment for 14 days, with the mass fraction of 2.5, 5, 7.5, 10, 12.5% BC and β-CD-BC added. To explore the effect of curing time, 10% PC, 30% deionized water, and 10% BC were added and fixed in a 24 °C incubator for 5, 10, 20, 30, and 40 days, respectively. The evaporated water was supplemented by a weighing method every three days.

Using the Box–Behnken design model of design-expert software, the selection of factors follows the principle of influence on as leaching (reagent ratio and curing time). In this study, response surface factor level experiments (Table 2) were used for the orthogonal design (Table 3). Taking BC + PC compound as the control, the key parameters such as optimal β-CD-BC + PC compound ratio and curing time were explored.

The leaching concentration of HMs was determined according to the standard toxic leaching procedure TCLP (Toxic Characteristic Leaching Procedure) recommended by the U.S. Environmental Protection Agency. The leaching rate of HMs was calculated according to the following formula:(1)$$\eta =\frac{c_{0}-c_{t}}{c_{0}}\times 100\%$$
where: *η* is the leaching rate; c_0_ is the leaching concentration of as in the original soil; c_t_ is the leaching concentration after solidification and stabilization.

#### 2.3.2. Unconfined Compressive Strength (UCS) Test

The procedure of UCS tests was according to ASTM D2166-91. Then, the specimen was placed on the bottom plate of test setup. The wheel was then turned to lower down the upper plate. When the upper plate was in contact with the top surface of the specimen, the indication of dial gauge was adjusted to zero. Then, the wheel was turned automatically at a speed of 0.06 mm/min (corresponding to 0.5% strain of dial gauge) to compress the specimen to failure.

#### 2.3.3. Analytical Method

The surface morphology of BC and β-CD-BC was observed by SEM (Sigma 300, Zeiss, Jena, Germany). The mineral composition of β-CD-BC was analyzed by X-ray diffraction (Ultima IV, Rigaku, Tokyo, Japan). The total arsenic was determined by the aqua regia perchloric acid method, and the metal speciation procedure applied for HMs was recommended by Liang et al. [19]. The data were processed by design expert.8.0.6.1, Excel2003, and Origin 8.0 software.

## 3. Results and Discussion

### 3.1. Characterization of BC and β-CD-BC

The Figure 2a–c shows the scanning electron microscope images of BC and β-CD-BC. The modified BC still retains the rich pore structure of BC, and the morphology of BC itself has not changed. White matter appears on the surface of β-CD-BC. The white point is more uniformly dispersed through images.

Shown in Figure 2b,c are β-CD-BC adsorbing HMs. It can be seen that there is no obvious change in the surface structure of BC adsorbed with HMs at different scales. As shown in the Figure 2d EDS analysis, the effect of β-CD-BC adsorbing HMs is obvious, and there are a large number of heavy metals on the surface of β-CD-BC. It can be considered that the β-CD did not agglomerate in a large amount, and the modification was relatively successful. Meanwhile, after β-CD functionalization BC lost its well-aligned structure, and a wealth of white substances appeared and covered the surface and pores of BC [20].

Figure 3 shows FTIR spectra that were created for evaluating the functional groups of β-CD, BC, and β-CD-BC. The influential band for −OH stretching in alkyl or aryl at 3434 cm^−1^, the bands at around 1631, 1030 cm^−1^ of BC can be assigned to the C-O stretching or O-H banding, and the band is identified for −CH at 608 cm^−1^. Additionally, β-CD-BC has similar analogous functional groups. The new bands at 1630 and 1460 cm^−1^ represent the COO- and CeN, respectively. Moreover, in comparison with pristine β-CD, the new band at around 608 cm^−1^ of BC could be indorsed to COO- stretching vibration. It could also be observed in β-CD-BC [21], confirming the introduction of the carboxyl groups onto the modified adsorbent. All the results proved that the BC had been successfully functionalized by β-CD [22].

It can be seen from Table 4 that the specific surface area of the BC modified by β-CD increased a lot, indicating that there is no accumulation during the modification process, and the β-CD is relatively uniformly dispersed on the surface of the BC. Additionally, the pore volume of β-CD-BC was higher than BC, while the average pore size of β-CD-BC was lower than BC. When subjected to β-CD, changes in surface properties are accompanied by changes in structure and composition, such as cation exchange capacity (CEC). The CEC of β-CD-BC is higher than that of BC (Table 3). The difference between the two functional BC is mainly related to the surface functional groups.

### 3.2. Solidification/Stabilization Experiment

#### 3.2.1. Single Factor Experiment

##### Leachability of PHe

As shown in Figure 4a, BC and β-CD-BC were added to the soil for maintenance at the mass ratios of 2.5%, 5%, 7.5%, 10%, and 12.5%. The leaching rate of PHE in the cured BC/PC samples was higher than that in the blank group, which may be due to the hydrophilic environment of PC, which increased the leaching concentration of PHH in the soil. With the increase of β-CD-BC, the leaching concentration of PHe in the soil gradually decreases until it becomes stable, because the cyclodextrin cavity on the surface of BC is hydrophobic and has a good adsorption effect on PHe. Through the above experiments, β-CD-BC can effectively reduce the leaching concentration of PHe in the soil and has a good stabilizing ability. As shown in Figure 4b, the PHe leaching concentration of the sample cured by β-CD-BC + PC reached a stable level after 20 d.

##### Leachability of Heavy Metals

As shown in Figure 5, BC and β-CD-BC were added to the soil for curing according to the mass ratios of 2.5%, 5%, 7.5%, 10%, and 12.5%. The leaching concentration of HMs in BC + PC samples was higher than that in the blank group. The oxygen-containing functional groups on the surface of BC can chelate and cooperate with HMs in soil, to reduce the leaching concentration of HMs in soil. BC has developed a pore structure and can adsorb HMs on the surface. The leaching concentration of HMs in the β-CD-BC + PC cured sample is lower than that of the blank group and BC + PC. This is because the hydroxyl and carboxyl functional groups on the surface of BC are modified by β-CD, which greatly improves the performance of the material. The chelation between HMs and hydrophilic functional groups enhances their stability. Meanwhile, PC can change the pH value of the soil, increase the content of -OH in the soil, and cause the co-precipitation of HMs and -OH [23]. Therefore, β-CD-BC significantly enhances the adsorption capacity of HMs.

#### 3.2.2. The Response Surface Experiments

##### The Results of Box–Behnken Design and the Variance Analysis of Model

The fitting equation of PHe is Y = 0.12 − 0.028A − 0.041B + 0.023C − 2.5 × 10^−3^AB − 0.016AC + 5 × 10^−3^BC + 0.011A^2^ + 0.022B^2^ − 0.012C^2^

Curing time (10−30d), Portland cement (6–18%), and BC (5–10%) are three reaction parameters that affect the leaching concentration of PHe. The results of ANOVA are shown in Table 5. The results of variance analysis show that the *p* values of β-CD-BC, PC, and curing time are less than 0.05, which indicates that the fitting effect of the model is good, and the response surface approximation model can be used to optimize the leaching concentration of heavy metal and PHe. According to the results of the analysis of variance in Table 5, the *p* values of A, B, C, and AC are all less than 0.05, indicating that curing time, BC ratio, and cement blending ratio are important factors for curing stability. It plays an important role in the leaching concentration of PHe. The *p* value of BC is the largest, which indicates that the effect of adding β-CD-BC and PC on the leaching concentration is relatively weak and has little impact on the final result. It also shows that cement enhances the leaching of some PHe and can be fixed by cyclodextrin on the surface of BC. The *p* values of the three influencing factors are 0.0012, 0.0002, and 0.0064, respectively. The order of influence on PHe leaching concentration is B (Biochar ratio) > A (Curing time) > C (Cement ratio).

The fitting equation of HMs is Y = 25.11 − 2.42A − 8.75B − 5.96C + 3.47AB − 0.62AC + 2.44BC + 4.06A^2^ − 1.03B^2^ + 2.11C^2^.

According to the results of variance analysis in Table 6, the *p* values of A, B, and C are all less than 0.05, indicating that the addition of β-CD-BC, PC, and curing time are essential factors in the solidification and stabilization process of HMs, which play a decisive role in the leaching concentration of heavy metals. The influence order of the three factors on HMs leaching is B (Biochar ratio) > C (Cement ratio) > A (curing time). The leaching concentration of heavy metals decreases with the increase of curing time under the condition of using the same proportion of reagents in comparison samples 1, 10, 2, 11, 3, 12, and 4, 13. Compared with samples 5–9, under the condition of curing for 20 days, the leaching concentration of HMs decreased with the increase of PC and BC, respectively.

As shown in Figure 5a,b, after curing for 7 days, the leaching amount of samples 1, 2, 3, and 4 increased with the increase of cement and BC amount. When the leaching agent is mixed with the sample, materials with lower cement content may form agglomerates, thereby reducing the contact area between the phases. In contrast, samples with higher cement content do not form clumps, thereby maintaining a higher interface contact area and forming a more hydrophilic environment, so the contact between the leachate and the stabilized/cured sample is better.

As is shown in Figure 6a, because β-CD-BC has hydrophobic interaction sites it can adsorb PHe well. When the curing time is about 14 days, the concentration of sample 6 is lower because less PC is added, and β-CD-BC can adsorb with PHe and reduce the concentration in the leachate. The higher concentration of sample 7 compared with sample 11 is due to the increased PHe leaching caused by the environment provided by cement, β-CB BC may not have enough time and PHe suction. Sample 13 contains more PC, which increases the concentration of PHe in the leachate, and the content of cyclodextrin in β-CD-BC is less, which provides fewer hydrophobic sites and cannot adsorb PHe. As is shown in Figure 6b, because BC does not have hydrophobic interaction sites it cannot adsorb PHe well. With the increase of PC content, the hydrophilic environment is caused, and the hydrophobicity of the sample decreases with the increase of the amount of hydrophilic cement, which increases the leaching concentration of PHe. The adsorption capacity of PHe is low only by π-π interaction.

It can be seen from Figure 7 that under the condition of the same reagent addition ratio, the HMs leaching concentration of comparison samples 1 and 10, samples 2 and 11, samples 3 and 12, samples 4 and 13 decreases with the increase of curing time. When the control samples were cured for 20 days, the HMs leaching concentration decreased with the increase of PC and BC. With comparing (a) and (f), (b) and (g), (c) and (H), (d) and (I), respectively, (e) and (J), it can be found that the leaching concentration of HMs treated with β-CD-BC is lower than BC.

##### Response Surface Analysis and Experimental Verification

Response surface analysis of PHe:

Figure 8a shows that the contour lines of the A and B factors present an oval shape, indicating an interaction between the two factors. Reasonable use of the relationship between A and B can effectively reduce the leaching concentration of PHe. Figure 8a shows that from the relationship between β-CD-BC and curing time, the response surface seen in the B direction is steeper and the contour lines are denser, which indicates that the effect of β-CD-BC and BC on the leaching concentration of PHe is more significant than that of curing time. Figure 8b shows the relationship between PC and curing time. It is seen that the response surface in A and C directions tends to be flat, indicating that curing time and PC dosage have the same effect on the leaching concentration of PHe. In Figure 8c the relationship is shown between the content of β-CD-BC and PC; it is seen that the response surface of B direction tends to be flat, but the contour lines of the two are oval, which indicates that the interaction between the two is vital. This is because PC provides a relatively hydrophilic environment for the soil, so that PHe in the soil precipitates, which is adsorbed on the β-CD-BC surface by cyclodextrin through hydrophobicity. Through the analysis of *p* value and response surface, the relationship between β-CD-BC and PC addition should be given priority.

Response surface analysis of heavy metals: As can be seen from Figure 9a,b, the contour is elliptical, indicating a specific interaction between maintenance time and the amount of BC added. With the extension of curing time, HMs were adsorbed in soil. The contour line in Figure 9b does not show an apparent ellipse, indicating that the interaction between BC and curing time is fragile. The relationship between β-CD-BC and curing time can be obtained from Figure 9a. It can be seen that the response surface in direction B is steeper and the contour line is denser. It shows that the effect of β-CD-BC addition on the leaching concentration of HMs is more significant than that of curing time. Increasing the amount of BC in the soil can better solidify and stabilize HMs. From Figure 9b, the relationship between PC and curing time can be obtained. It can be seen that the response surface in C is steep, while the response surface in direction a tends to be flat, which indicates that PC has a more significant effect on the leaching concentration of HMs than PC, indicating that PC adsorbs and precipitates HMs. Through the analysis of *p* value and response surface, when optimizing the factors, priority should be given to β-CD-BC and PC.

Verification test: The results show that the curing time is 22.85 d, β-CD-BC content of 9.75% and PC content of 11.4% are the best parameters for curing stability. To verify the reliability of the model, according to the optimal parameter combination, the theoretical leaching concentration of HMs in soil was as low as 16.81 mg/kg and that of PHe was as low as 0.054 μg/kg, which is close to the predicted value. The results show that the optimization method and model are reliable.

Figure 10 shows speciation characteristics of HMs. The changes of Cr, Cd, Cu, Pb, and Zn in the soil after adding BC and β-CD-BC + PC are shown in Figure 10. The proportion of Cd residual in the untreated soil was 37%, while the proportion of Cd treated with BC + PC and β-CD-BC + PC was 65% and 73%, respectively. Meanwhile, the weak acid soluble state and reducible state of soil Cd treated with BC + PC and β-CD-BC + PC decreased from 13% and 9% to 3–6% and 4–7%, respectively. The re sidual state of Cr, Cu, Pb, and Zn in the untreated soil was reduced compared with the soil treated with BC + PC and β-CD-BC + PC, respectively. At the same time, the weak acid extraction and reducible states of Cr, Cu, Pb, and Zn in the treated soil were reduced. This is because the added BC and β-CD-BC are adsorbed with HMs, and the surface of the material contains surface functional groups (-OH, -COOH, etc.) that can undergo complex chelation. The PC added to the soil can not only immobilize HMs in the soil, but also change the pH of the soil, causing HMs to precipitate, thereby reducing mobility. It can be found from Figure 10 that the ratio of Cr, Cu, Pb, and Zn in the residue state of the soil treated with β-CD-BC is lower than that of the residue state in BC/CP. The weak acid soluble state of metals in the soil treated with β-CD-BC is the lowest, because a large amount of cyclodextrin is attached to the surface of the modified BC, and the surface of the cyclodextrin cavity has a hydrophilic effect and contains a large number of hydroxyl groups. It can complex with HMs in the soil, change the form of HMs, and produce better stabilization effects on these.

### 3.3. Synchronous Adsorption Investigations of Pb, Cu, Cr, Cd, Zn, and PHe System

As shown in Figure 11, there is synchronous adsorption of Pb, Cu, Cr, Cd, Zn, and PHe. To realize the simultaneous solidification and stabilization of HMs and PHe in soil, BC was modified to have the hydrophobic properties of β-CD. Meanwhile, the oxygen-containing functional groups on the surface of BC increased and complexed with HMs, which enhanced the curing and stability of the material. After modification, the leaching concentration of Pb, Cu, Cr, Cd, and Zn is reduced, and the curing and stabilizing effect on PHe is also better. The main reason is that Pb, Cu, Cr, CD, Zn, and PHe have different absorption mechanisms in the soil. The cyclodextrin of β-CD-BC can wrap PHe in the cavity of the adsorbent through hydrophobic interaction, and the aromatic structure on the surface of the BC can provide potential π-π interaction [24,25] and PHe accumulation, which enhances the adsorption of PHe [26]. The adsorption of Pb, Cu, Cr, Cd, and Zn is related to the complexation and electrostatic interaction between the oxygen-containing groups of β-CD-BC [27]. The developed pore structure on BC surface can cause the surface adsorption of HMs. Meanwhile, PC can change the pH of the soil, causing the precipitation of HMs and -OH in the soil [23].

In short, as shown in Figure 11, β-CD-BC achieves the adsorption of PHe through hydrophobic effect and π-π interaction, and enhances the adsorption of Pb, Cu, Cr, Cd, and Zn through complexation and electrostatic attraction [28]. This avoids the competition between the two pollutants for adsorption sites and improves the adsorption performance and adsorption capacity of BC.

### 3.4. Unconfined Compressive Strength (UCS) Test

The UCS of the samples are given in Figure 12. It can be seen from Figure 12 that the compressive strength value of the cured body with the same addition of 6% of samples 1, 6, 11, and 16 shows a trend of increasing first and then being stable with the increase of curing time. When the curing time is 7 days, the compressive strength of the solidified sample1 is 33 kPa. The compressive strength value of the solidified sample is lower than the detection limit, which cannot meet the solidification requirements. This may be due to the relatively high water content in the solidified sludge, which results in the dispersion of cement particles and reduces the hydration reaction [29]. The cementing ability of the generated hydrated calcium silicate makes the compressive strength of the solidified sample low. As the curing time increases to 14 d, 21 d, 28 d, the compressive strength of the cured body increases to 52–57 kPa, adding 12% of No. 2, 5, 12, and 15 samples, adding 18% of No. 3, 8, 9, 14 samples, and adding 24% of No. 4, 7, 10, and 13 samples. With the increase of curing time, the compressive strength of solidified body first increases and then stabilizes. The compressive strength of the sludge solidified body reaches the maximum when the solidification time is 21 d [30].

In the process of curing and stabilization, the compressive strength of the cured body is positively correlated with the curing time. The results show that the selection of reasonable curing time and cement mix proportion is the key to the sludge curing process.

## 4. Conclusions

(1)The number of oxygen-containing functional groups on the surface of BC modified by β-CD increased. β-CD-BC still has well-developed void structure and specific surface area.(2)The compressive strength test shows that the strength is positively correlated with the amount of PC and the curing time, and the compressive strength is stable after 14 days. After β-CD-BC/PC curing, the speciation of HMs changed significantly, the residual Cu content increased from 56% to 75%; the residual Cr content increased from 54% to 82%; the residual CD content increased from 37% to 77%; residual Zn content increased from 61% to 87%; residual Pb increased from 59% to 79%.(3)β-CD-BC achieves the adsorption of PHe through hydrophobic effect and π-π interaction, and enhances the adsorption of Pb, Cu, Cr, Cd, Zn through complexation and electrostatic attraction. This avoids the competition between the two pollutants for adsorption sites and improves the adsorption performance and adsorption capacity of BC.

## Figures and Tables

**Figure 1 ijerph-19-01060-f001:**
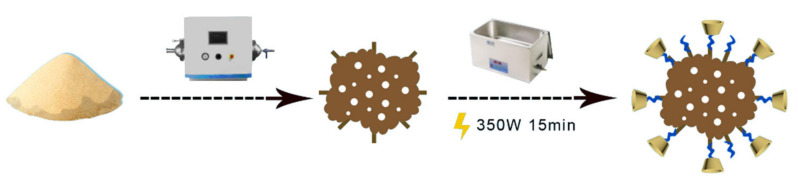
β-CD-BC preparation process.

**Figure 2 ijerph-19-01060-f002:**
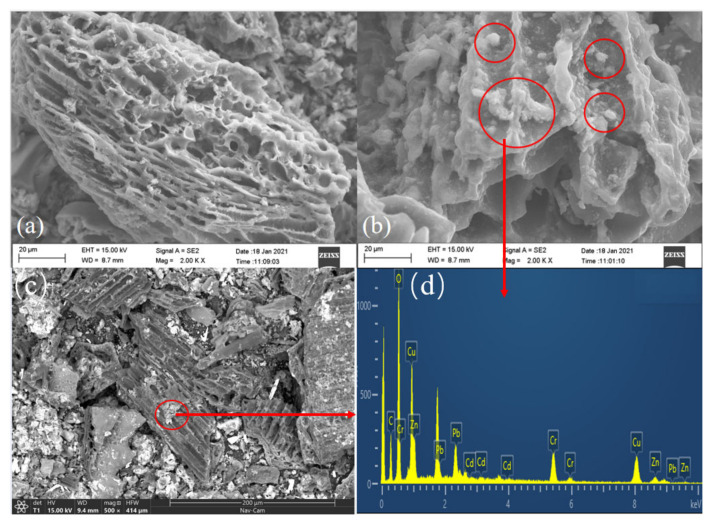
SEM images, (**a**) BC, (**b**) β-CD-BC, (**c**) β-CD-BC.EDS images, (**d**) β-CD-BC.

**Figure 3 ijerph-19-01060-f003:**
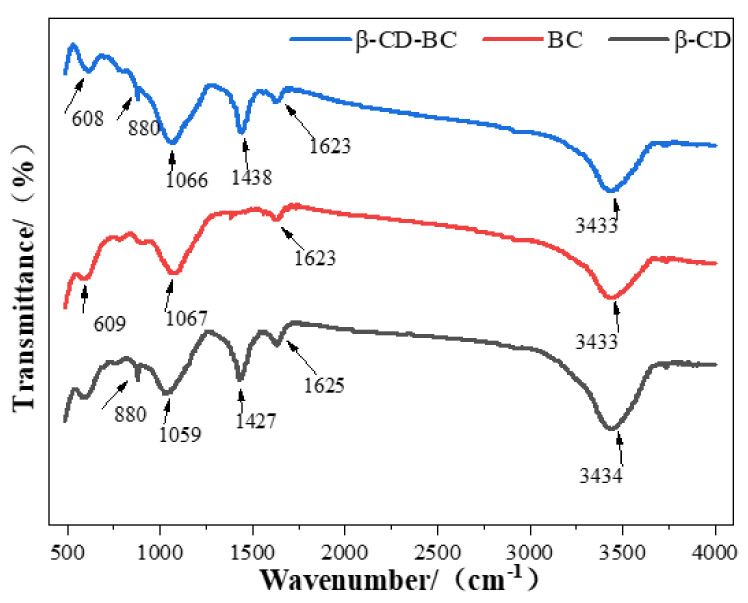
FTIR spectra of BC and β-CD-BC.

**Figure 4 ijerph-19-01060-f004:**
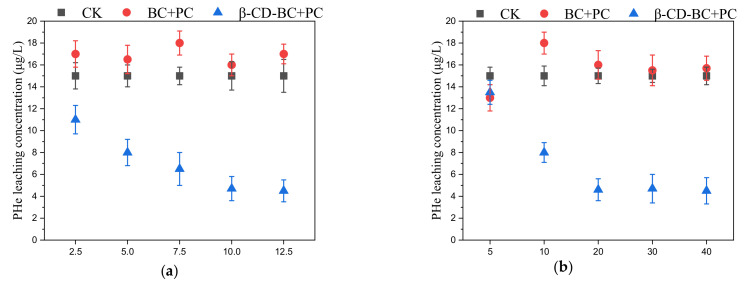
Effects of different BC and β-CD-BC (**a**) addition amounts and (**b**) curing times on PHe leaching concentration.

**Figure 5 ijerph-19-01060-f005:**
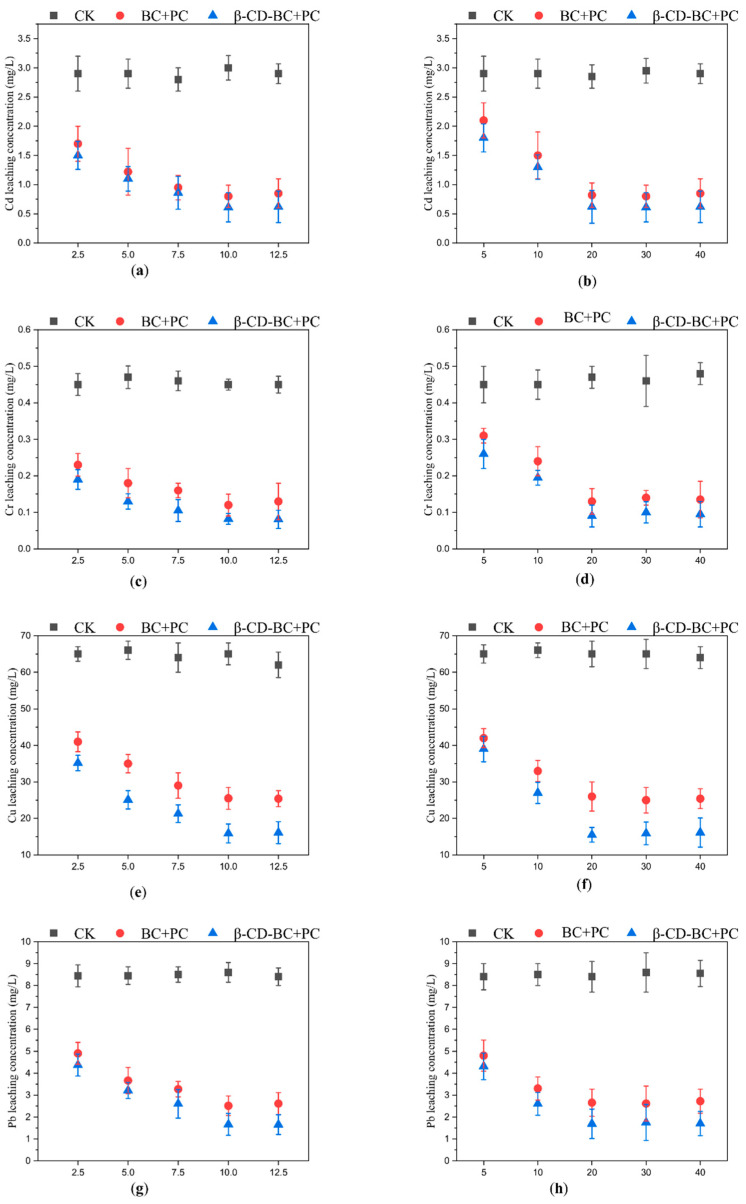
Effects of different BC and β-CD-BC addition amounts and curing times on HMs leach-ing concentration. (**a**) Addition amount and (**b**) Curing time for Cd leaching concentration; (**c**) Addition amount and (**d**) Curing time for Cr leaching concentration; (**e**) Addition amount and (**f**) Curing time for Cu leaching concentration; (**g**) Addition amount and (**h**) Curing time for Pb leaching concentration; (**i**) Addition amount and (**j**) Curing time for Zn leaching concentration.

**Figure 6 ijerph-19-01060-f006:**
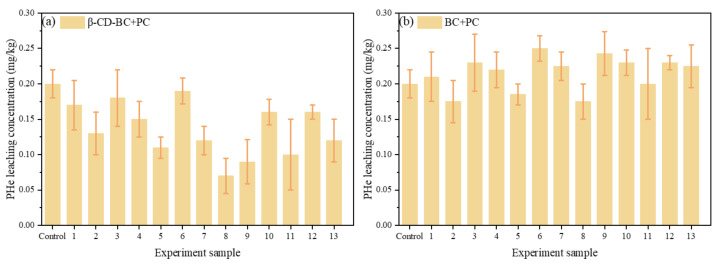
Leaching concentration of PHe in Box–Behnken design for (**a**) β-CD-BC + PC and (**b**) BC + PC.

**Figure 7 ijerph-19-01060-f007:**
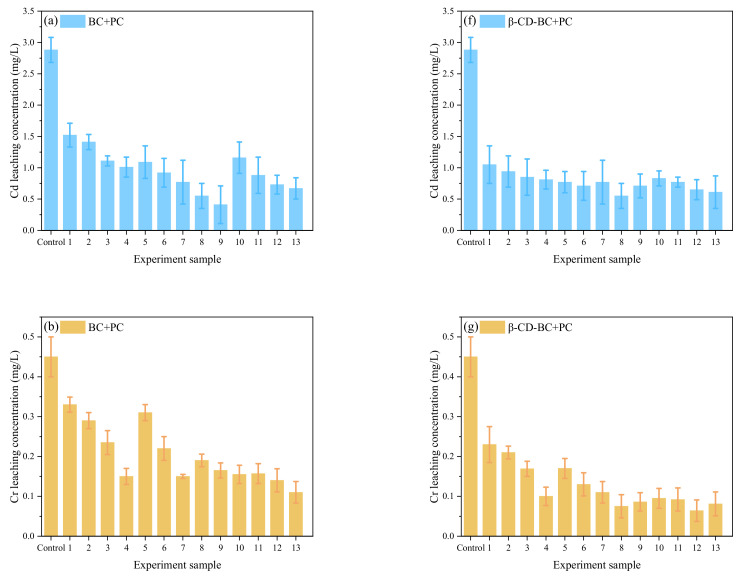
Leaching concentration of HMs in Box–Behnken design. Cd leaching concentration of (**a**) BC + PC and (**f**) β-CD-BC + PC; Cr leaching concentration of (**b**) BC + PC and (**g**) β-CD-BC + PC; Cu leaching concentration of (**c**) BC + PC and (**h**) β-CD-BC + PC; Pb leaching concentration of (**d**) BC + PC and (**i**) β-CD-BC + PC; Zn leaching concentration of (**e**) BC + PC and (**j**) β-CD-BC + PC.

**Figure 8 ijerph-19-01060-f008:**
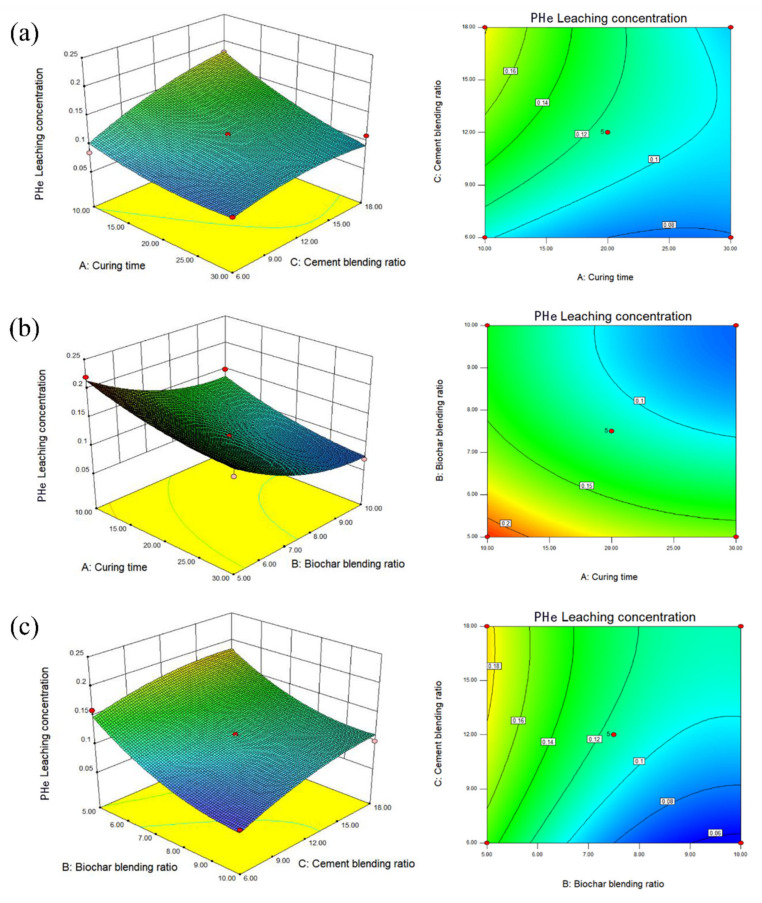
Leaching concentration of PHe. (**a**) Curing time and biochar blending ratio. (**b**) Biochar blending ratio and cement blending ratio. (**c**) Curing time and cement blending ratio.

**Figure 9 ijerph-19-01060-f009:**
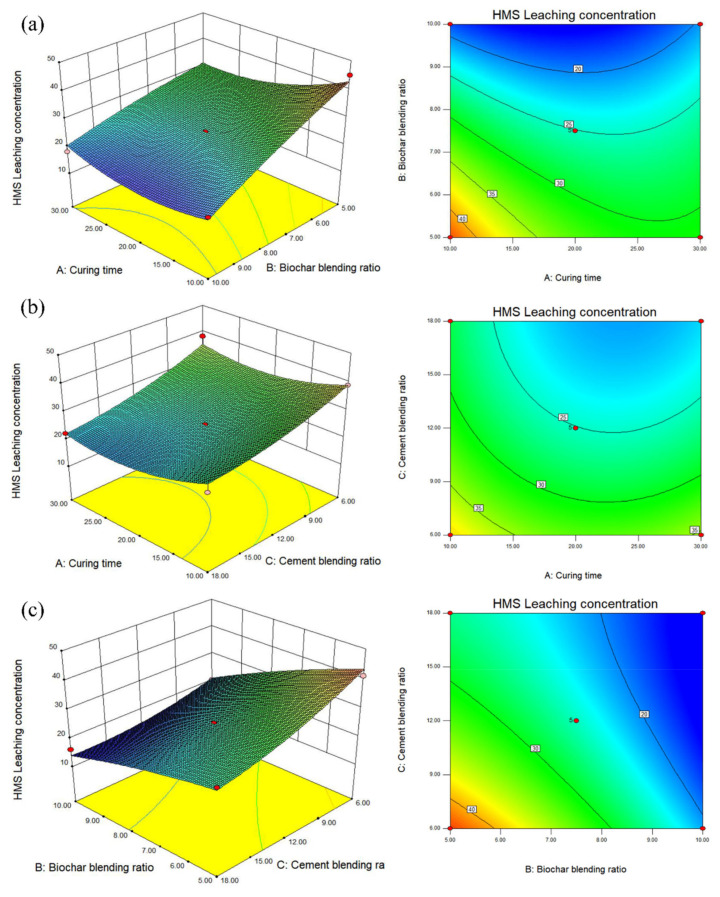
Leaching concentration of HMs. (**a**) Curing time and biochar blending ratio. (**b**) Biochar blending ratio and cement blending ratio. (**c**) Curing time and cement blending ratio.

**Figure 10 ijerph-19-01060-f010:**
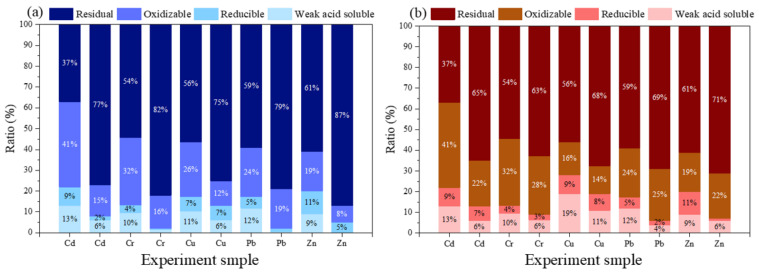
Speciation characteristics of HMs, (**a**) β-CD-BC + PC, (**b**) BC + PC.

**Figure 11 ijerph-19-01060-f011:**
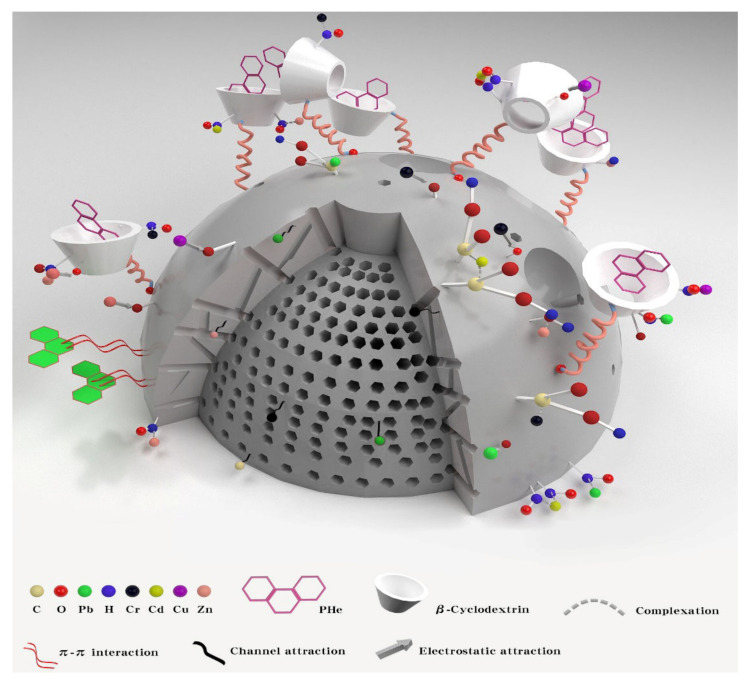
Synchronous adsorption of Pb, Cu, Cr, Cd, Zn, and PHe.

**Figure 12 ijerph-19-01060-f012:**
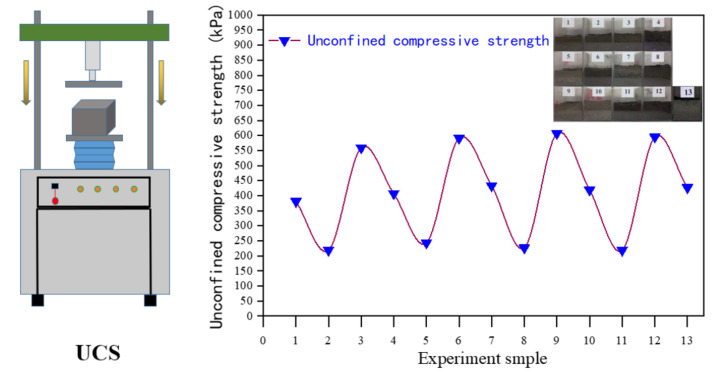
Unconfined compressive strength of the solidified specimen.

**Table 1 ijerph-19-01060-t001:** Physical and chemical properties of soil.

pH	Organic Matter/g·kg^−1^	Particle Composition/%			TN/g·kg^−1^	TP/g·kg^−1^	CEC/cmol·kg^−1^
		Sand	Power	Clay			
7.6	26.31	26.86	49.41	25.12	1.05	6.53	12.6

**Table 2 ijerph-19-01060-t002:** Box–Behnken factors and levels of experimentation.

Specimen	Curing Time (d)	Biochar Ratio (%)	Cement Ratio (%)
1	10	5	6
2	20	7.5	12
3	30	10	18

**Table 3 ijerph-19-01060-t003:** Box–Behnken experimental design table.

Sample	A Curing Time (d)	B Biochar Ratio (%)	C Cement Ratio (%)
1	10	5.0	12
2	10	7.5	6
3	10	7.5	18
4	10	10.0	12
5	20	5.0	6
6	20	5.0	18
7	20	7.5	12
8	20	10.0	6
9	20	10.0	18
10	30	5.0	12
11	30	7.5	6
12	30	7.5	18
13	30	10.0	12

**Table 4 ijerph-19-01060-t004:** Surface characteristics of biochars.

Specimen	Specific Surface Area	Pore Volume (cm^3^/g)	Average Pore Size (nm)
β-CD-BC	115.3880	0.141730	4.91317
BC	51.3219	0.077551	4.91317

**Table 5 ijerph-19-01060-t005:** Analysis of variance for response surface quadratic mode of PHe.

Source	df	Sum of Squares	Mean Squares	F Value	Prob > F
Model	9	2.6 × 10^−2^	2.883 × 10^−3^	12.47	0.00016
A	1	6.328 × 10^−3^	6.328 × 10^−3^	27.36	0.0012
B	1	1.200 × 10^−2^	1.200 × 10^−2^	51.95	0.0002
C	1	3.403 × 10^−3^	3.403 × 10^−3^	14.72	0.0064
AB	1	2.500 × 10^−5^	2.500 × 10^−5^	0.11	0.7519
AC	1	1.056 × 10^−3^	1.056 × 10^−3^	4.57	0.0699
BC	1	0.000	0.000	0.000	1.0005
A2	1	2.780 × 10^−4^	2.780 × 10^−4^	1.20	0.3092
B2	1	2.502 × 10^−3^	2.502 × 10^−3^	10.82	0.0133
C2	1	3.701 × 10^−4^	3.701 × 10^−4^	1.60	0.2463
Residual	7	1.619 × 10^−3^	5.396 × 10^−4^		

**Table 6 ijerph-19-01060-t006:** Analysis of variance for response surface quadratic mode of HMs.

Source	df	Sum of Squares	Mean Squares	F Value	Prob > F
Model	9	111.10	123.46	21.50	0.0003
A	1	46.80	46.80	8.15	0.0245
B	1	612.15	612.15	106.60	<0.0001
C	1	284.53	284.53	49.55	0.0002
AB	1	48.02	48.02	8.36	0.0233
AC	1	1.55	1.55	0.27	0.6194
BC	1	23.81	23.81	4.15	0.0811
A2	1	69.36	69.36	12.08	0.0103
B2	1	4.46	4.46	0.78	0.4076
C2	1	18.81	18.81	3.28	0.1132
Residual	7	40.20	5.74		

## Data Availability

Data sharing not applicable. No new data were created or analyzed in this study. Data sharing is not applicable to this article.

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
