# Peer review of "Stabilization/Solidification of Heavy Metals and PHe Contaminated Soil with β-Cyclodextrin Modified Biochar (β-CD-BC) and Portland Cement"

_ijerph, 2022, doi:10.3390/ijerph19031060_

Round 1

Reviewer 1 Report

There are technical and non technical comment for the author. Author should avoid some mistakes and can improve the manuscript as per the following comments:

  • Abstract should be more informative and it reflects some more information carrying in the rest of manuscript.
  • Why only Cr, Cd, Cu, Pb, and Zn have been selected for treatment. As there are many other heavy metals are too present in soil.
  • In the manuscript, I have not seen the explanation and information of Figure 11 and Figure 12. Authors should be suggested to incorporate some information of Figure 11 and 12. In the same way, Figure 1, Figure 2 and Figure 3 have not been mentioned in the manuscript text. Do include these figures and mention the in the text.
  • On line number 350, Is the mentioning of Fig 10 is correct? or this is the information of Figure 11. Author are suggested to check these types of mistake for Figure 10,11 and 12.
  • The Table 1 has been included in the manuscript. The title of this Table is missing. Do mention the use of this Table in the manuscript. Give this Table reference in the manuscript.
  • On the line number 315 and 326, Do mention the figure numbers.
  • If possible, Do include one /two more SEM images at different resolutions. Energy-dispersive X-ray spectroscopy (EDX) can be added to know the exact weight percentage.
  • Along with FTIR, authors can include XRD studies to increase the better understanding of material.
  • Compressive strength of sample contaminated with different ranges (such as 500 mg/kg, 1500 mg/kg) of heavy metal ions can be used in S/S after immersion for different time series of day and graphs can be provided for the same.
  • There is only one reference form 2021. Author should be advised to add some more recent reference of 2021.

Reviewer 2 Report

Manuscript Number: ijerph-1494874

Title:Stabilization/solidification of Heavy meatls and PHe contaminated soil with β-Cyclodextrin modified biochar ï¼ˆβ-CD-BC)and Portland cement

Authors: Gen Li, Haibo Li, Yinghua Li, Xi Chen, Xinjing Li, LixinWang, Wenxin Zhang, Ying Zhou

I consider that these data presented in this manuscript may be useful, having in mind their impact (e.g., ambiental and biological). However, some major points must be re-considered by the authors. Therefore, publication after major revision is suggested by this reviewer.

The specific comments are shown below. That is,

  1. The language needs reorganizing throughout the manuscript to make the paper brief and accurate. (e.g., Title, Abstract and Experimental section).
  2. In section 2.1 Materials and reagents, the authors must insert the mass fraction purity and the CAS Number of each reagent used. In fact, the impurities can influence the properties of these compounds. In addition, why the authors did not dry the respective salts, such as recommend by C. Duval in his book (i.e., Inorganic Thermogravimetric Analysis, Elsevier 1953), or, why did not made a characterization using a thermogravimetric analysis?
  3. In addition, it is not clear what kind of water did they use for these measurements. I think it would be useful to indicate the specific conductivity of the used water.
  • The meaning of symbols in all the manuscript (text and figures) should be indicated. (e.g., PHe in Figures 4 and 5)
  • In figure 3, the authors must write Wavenumber/(cm-1)instead Wavenumber(cm-1)IUPAC Rules). I recommend the authors to see this point in all manuscript, including tables, figures and the text.
  • The scientific rigor in this kind of work is sometimes absent. For example, on page 8, I recommend to review the fitting equation of PHe The authors are invited to clarify the meaning of symbols and to present values of R2, or other or other adjustment parameter. I recommend to review all fitting equations here indicated.
  • Uncertainties in all measured variables must be given.

In conclusion, the scientific rigor in this kind of this manuscript is frequently absent and, consequently, I am unable to accept the paper in its present form. Consequently, having in mind these points, I recommend reviewing all the text.

Round 2

Reviewer 2 Report

Dear Editor,

The authors were careful to answer all the questions raised by the referees, as well as making changes to the text, resulting in an improved manuscript. Consequently, I consider that this paper can be published in the International Journal of Environmental Research and
Public Health

My Compliments